# Evaluation of the Environmental Burden of Cross-Laminated Timber Manufacturing in Japan Using the Input–Output Analysis

Mengyuan Liu [1],*[ID], Akito Murano [2], Chun Sheng Goh [3,4] and Chihiro Kayo [5]

1 United Graduate School of Agricultural Science, Tokyo University of Agriculture and Technology, 3-5-8 Saiwai-cho, Fuchu, Tokyo 183-8509, Japan
2 Faculty of Science and Engineering, Toyo University, 2100 Kujirai, Kawagoe, Saitama 350-8585, Japan; amurano@toyo.jp
3 Harvard University Asia Center, Center for Government and International Studies (CGIS) South, 1st Floor 1730 Cambridge Street, Cambridge, MA 02138, USA; chunshenggoh@fas.harvard.edu
4 Jeffrey Sachs Center on Sustainable Development, Sunway University, No. 5, Jalan Universiti, Bandar Sunway, Subang Jaya 47500, Selangor Darul Ehsan, Malaysia
5 Institute of Agriculture, Tokyo University of Agriculture and Technology, 3-5-8 Saiwai-cho, Fuchu, Tokyo 183-8509, Japan; kayoc@cc.tuat.ac.jp
* Correspondence: s217129y@st.go.tuat.ac.jp

**Abstract:** Japan is actively promoting the application of cross-laminated timber (CLT) in construction to utilize plantation forests efficiently and fulfil its climate commitments. Although CLT has unique structural properties and environmental advantages, understanding the environmental burden of CLT manufacturing remains scarce. This study uses input–output analysis to evaluate the greenhouse gas (GHG) emissions from CLT manufacturing. An extended input–output table was created to measure the GHG emissions by investigating the revenue and expenditure data of the largest CLT manufacturers in Japan in 2020, combined with the energy and emission intensity data. The results showed that electricity, activities not elsewhere classified, road freight transportation (except self-transportation), timber, and logs were the main sectors contributing to GHG emissions from CLT manufacturing. In addition, the environmental burdens of the cement and steel sectors were evaluated for comparison with the same increase in the final demand. We found that CLT manufacturing emits significantly fewer GHGs than the cement and steel sectors. These findings highlight the potential of CLT in reducing environmental burden, particularly in construction and civil engineering, emphasizing the importance of renewable energy use and efficient raw material transportation.

**Keywords:** cross-laminated timber; environmental burden; greenhouse gas emissions; energy consumption; input–output analysis; Japanese timber industry; cement industry; steel industry

## 1. Introduction

The Paris Agreement, adopted in 2015, set the goal of limiting the increase in the global average temperature to 2 °C above preindustrial levels through reaching carbon neutrality by 2050 [1]. Reducing GHG emissions and increasing greenhouse gas removal globally is vital for achieving these goals. The UNEP Emissions Gap Report of 2022 indicates that the growth rate of GHG emissions has slowed in the past decade but could still reach a new record in 2021 [2]. The building and construction sector accounts for 35% of the global energy use and 38% of $CO_2$ emissions across all impact sectors [3]. The steel and cement used to construct buildings are considered to be carbon-emission-intensive materials [4,5]. Japan's Intended Nationally Determined Contribution (INDC) indicates that Japan aims to achieve at least a 50% reduction in global GHG emissions by 2050 (compared with those in 2013) [6].

In recent years, plantation forests in Japan have reached a harvestable stage. The wood self-sufficiency ratio has increased over 20 consecutive years, reaching 41.8% by 2020 [7,8]. To promote the full utilization of domestic forestry resources and contribute to climate commitment, the Japanese government published a policy in 2010 to promote the application of timber in public buildings. In that context, cross-laminated timber (CLT), is defined by Japanese Agricultural Standards (JAS) as a "a general timber is made of sawn timber or small square timber laminated or glued together with the fiber direction nearly parallel to each other in width direction, mainly laminated and glued with the fiber direction nearly orthogonal to each other, and have a structure of three or more layers" [9], and has attracted attention because of its low weight (approximately 1/6–1/4 that of concrete) and rigidity in two-direction structures for non-residential and mid-rise applications, and its production is expected to continue to increase [7,10–12]. The "Roadmap for the Diffusion of CLT", jointly released by the Ministry of Land, Infrastructure, Transport, and Tourism (MLIT) and Japan Forestry Agency, indicated that the production capacity of CLT will reach 500,000 $m^3$/year by 2024 [10]. In addition, CLT has emerged as a sustainable alternative to cement and steel in buildings because of its environmental benefits [13].

Previous studies on the environmental burden of CLT have predominantly focused on two main aspects. Firstly, studies have explored the environmental burden of CLT buildings, highlighting their reduced energy loss compared to reinforced concrete (RC) in life cycle assessments (LCA) and reduced global warming potential compared to glued laminated timber (GLT) [14–17]. Liu et al. conducted an LCA of seven-story buildings in China and found that CLT buildings had $CO_2$ emissions over 40% lower compared to those made of RC [18]. A similar study in the United States found that CLT structure building reduced GHG emissions by an average of 26.5% [19]. A study in Malaysia revealed that CLT buildings are 7% more expensive but reduce the embodied energy by 40% compared to GLT buildings [17].

Other studies analyzed the sources of GHG emissions within the manufacturing process to find ways to reduce environmental burden [20,21]. Nakano et al. surveyed key data from three major CLT manufacturing companies in Japan, and suggested reusing CLT panels over time to avoid the release of biocarbons into the atmosphere [20]. Chen et al. [21] studied five sawmills in western Washington, and found that the mill location and wood species mix were essential determinants of the environmental burden of CLT production. While these studies focused on understanding the environmental advantages of CLT application, a detailed assessment of the GHG emissions from the manufacturing stages and along the supply chain is required.

In our previous study, we quantified the economic impacts of CLT manufacturing in Japan by extending the input–output table. The preliminary study identified the activities not elsewhere classified, timber, logs, road freight transport (except self-transport), and wholesale trade sectors as the five sectors with the largest economic impacts [22]. Based on these results, we advanced our analysis using energy and emission intensity data, which allowed us to calculate the energy consumption and GHG emissions generated by the economic activities of CLT manufacturing and its supply chain. Therefore, we applied the economic results of the previous study and included the environmental burden intensities to calculate the energy consumption and GHG emissions of all economic sectors associated with CLT manufacturing. In addition, we compared the environmental burden from CLT manufacturing with that of cement and steel production for the same increase in final demand. This comprehensive approach provides new methodologies and data support for climate change responses and policymaking. By examining the environmental burden of the development of new wood industries, we aim to contribute valuable insight that can inform sustainable practices worldwide.

## 2. Target and Methods

### 2.1. Target CLT Manufacturer

In our previous study [22], we investigated the largest CLT manufacturer in Japan by production capacity, namely Meiken Lamwood Corp., located in Maniwa City, Okayama Prefecture. The company's business includes producing glued laminated timber (GLT) and CLTs, designing and constructing wooden buildings, and generating biomass power. The company has an annual production capacity of 30,000 $m^3$, producing CLTs up to $270 \times 3000 \times 12,000$ mm in size [23]. After 6 months of investigation, from October 2021 to March 2022, we obtained data on the manufacturer's revenues and expenditures for 2020. CLT production was 6399 $m^3$ during that period, and domestic final demand increased by JPY 923,868,000.

### 2.2. Evaluation Scope and Process Steps

This study assessed the energy consumption and GHG emissions of CLT manufacturing and its chain based on the results of preliminary economic ripple effects, combined with the 2015 Embodied Energy and Emission Intensity Data for Japan using input–output tables (3EID) developed by the National Institute for Environmental Studies (NIES) in Japan [22,24]. While input–output analysis was developed for economic studies, the generic input–output framework has been applied to environmental impacts analysis, including GHG emissions, land disturbance, and water and energy use [25–27]. The 2015 3EID includes the energy consumption and GHG emissions in each sector, as in the case of the national GHG inventory, raw material and fuel inputs for each sector were calculated based on comprehensive energy statistics from the Agency for Natural Resources and Energy (ANRE). Raw material and fuel inputs for each sector are dominated by non-renewable resources such as coal and coal products, petroleum and petroleum products, natural gas and municipal gas, etc., with a small proportion of renewable energy generation from water and nuclear energy [24]. Thus, in this study, the energy consumption comes from renewable and non-renewable resources, and GHG emissions come from non-renewable resources such as fossil fuels. Six target GHGs were fuel-derived and non-fuel-derived carbon dioxide ($CO_2$), methane ($CH_4$), nitrous oxide ($N_2O$), hydrofluorocarbons (HFCs), perfluorocarbons (PFCs), and sulfur hexafluoride ($SF_6$), which the Paris Agreement covers. However, emissions and removal from land use, land use change, and forestry (LULUCF) (emissions and removal involving forests, agricultural land, and other land uses) were not included in 2015 3EID [24]. We used producer price data from the 2015 3EID and merged 390 sectors into 187 sectors based on the 2015 input–output table sub-categories to correspond to economic and environmental data for the same sector [28].

The process of assessing environmental burden using the input–output table is as follows. First, an extended input–output table was created by adding a new CLT manufacturing sector from the 2015 input–output table, in which CLT manufacturing belongs to the "plywood, glued laminated timber (GLT)" sector (187 sectors) (Figure 1) [22,29]. The dashed lines in Figure 1 represent the flow of funds for revenues and expenditures of producing CLT, and solid lines represent the direction of inputs and outputs of the CLT manufacturing sector. It is worth mention that "Activities not elsewhere classified" is a kind of sector in 2015 Japanese input–output table, which covers the production activities of goods or services that are not elsewhere classified. And this sector also serves as the accumulation part of errors in the estimation of column and row sectors [28].

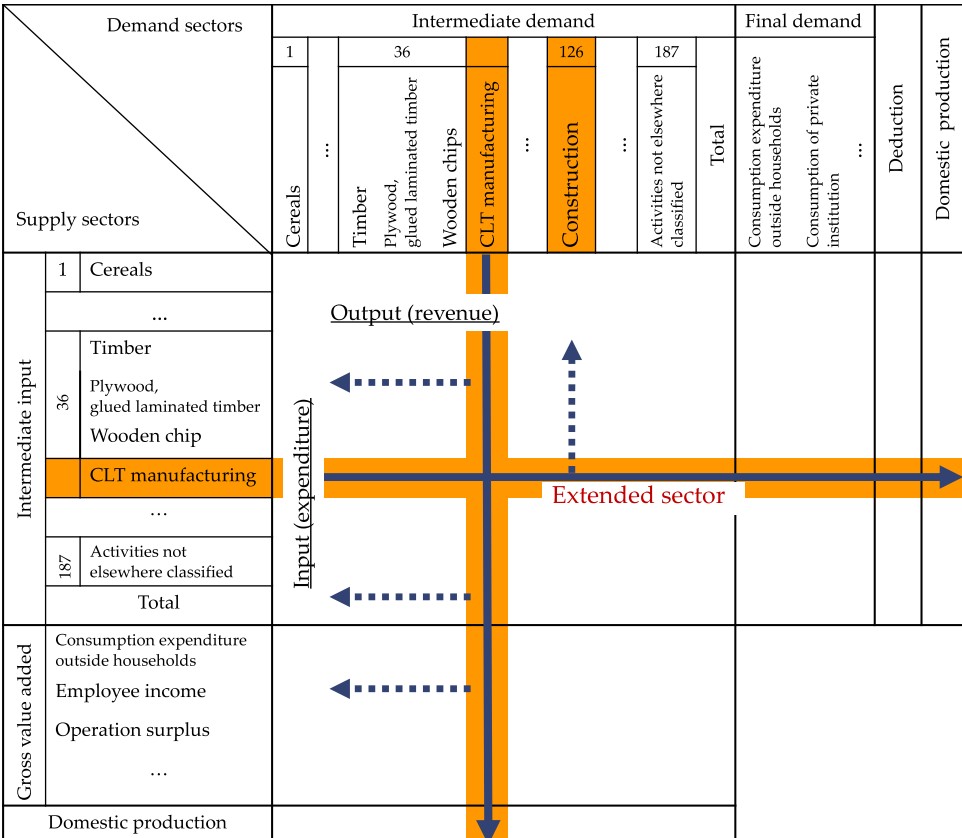

**Figure 1.** Schematic diagram of the extended input–output table [22]. (The extended sector is CLT manufacturing sector; the downward and leftward arrows indicate the inputs of CLT manufacturing to each sector; the rightward and upward arrows indicate the outputs of CLT manufacturing to each sector).

Then, the revenue and expenditure data were applied to the rows and columns of the CLT manufacturing sector. Based on the survey results, we combined residential and non-residential construction into the construction sector and attributed the revenue data to the construction sector. Expenditure data were then attributed to corresponding sectors (Table 1) [28]. Based on this extended basic transaction table, two tables were built: the input coefficient table and an inverse matrix coefficient table. This allowed the direct, indirect, induced, and total economic ripple effects to be calculated. Finally, direct, indirect (including indirect and induced), and total energy consumption and GHG emissions were calculated by combining the environmental burden intensities. Direct energy consumption and GHG emissions originate from increased production activities in sectors generated by final demand. Indirect energy consumption and GHG emissions originate from production activities caused by the direct economic ripple effects. Induced energy consumption and GHG emissions originate from the new consumption associated with employee revenue generated by direct and indirect economic ripple effects. We collectively refer to the indirect and induced energy consumption and GHG emissions as indirect energy and GHG emissions. Additionally, to provide a more visual comparison of GHG emissions and impacted sectors, we assessed the GHG emissions of the cement and cement product sectors (combined into the cement sector) and the pig iron and crude steel sectors (combined into the steel sector) simultaneously for the same increase in final demand.

**Table 1.** Assigning expenditure data to sectors [22].

| Materials and Expenditures | Sectors | |
| --- | --- | --- |
| | Intermediate Sector | Gross Value Added |
| Material cost (lamina) | Timber | |
| Material cost (adhesive) | Miscellaneous final chemical products | |
| Material cost (some transportation costs) | Road freight transport (except self-transport) | |
| Salary allowance | | |
| Excess work allowance | | Wages and salaries |
| Bonuses | | |
| Legal welfare expenses | | Consumption expenditure of households |
| Welfare expenses | | |
| Freight | Road freight transport (except self-transport) | |
| Consumable expenses | Office supplies | |
| Vehicle-related expenses | Petroleum refinery products 40%, machine repair services 60% | |
| Rental expenses | Goods rental and leasing (except car rental) | |
| Insurance expenses | Insurance | |
| Repair expenses | Machine repair services | |
| Fuel expenses | Petroleum products | |
| Utility expenses | Electricity 94%Steam and hot water supply 5%Water supply 1% | |
| Packaging | Packaging | |
| Tax and rent | | Indirect taxes (except custom duties and commodity taxes on imported goods) |
| Travel and transportation expenses | | Consumption expenditure of households |
| Depreciation and amortization | | |
| Small depreciable assets | | Consumption of fixed capital |
| Lump-sum depreciable assets | | |
| Storage charges | Storage facility service | |
| Communication expenses | Communications | |
| Business fees | Miscellaneous business services | |
| Security and cleaning expenses | | Wages and salaries |
| Recruitment and training expenses | | |
| Meeting expenses | Office supplies | |
| Allocation to other departments | | Wages and salaries |
| Expense transfer (dry steam, etc.) | Activities not elsewhere classified | |

*2.3. Constructing the Basic Transaction Table*

The revenue and expenditure data investigated were purchaser prices, including trade margins and domestic freight, whereas the input–output tables reflect the producer's prices. Thus, excluding trade margins and domestic freight from the revenue and expenditure data was necessary to unify producer prices. Trade margins included wholesale and retail, and domestic freight covered seven categories: railway, road, coastal, harbor, air, consigned

freight forwarding, and storage facility services. We used the output table (integrated sub-categories) to calculate the trade margin ratios and domestic freight ratios [30]. Producers' price data for revenue and expenditure were assigned to relevant sectors that manufactured the CLT. Excluded trade margins and domestic freight were sorted into the commercial and transportation sectors manufacturing the CLT. However, CLT manufacturing was included in the plywood GLT sectors in the 2015 input–output tables. Therefore, the input (expenditure) and output (revenue) data for the CLT manufacturing sector were excluded from the plywood GLT sectors to create an extended input–output table (basic transaction table) [22].

*2.4. Domestic Self-Sufficiency Ratios*

All wood utilized to manufacture CLT in this study was harvested, purchased, and processed domestically. Thus, the domestic self-sufficiency ratios for the logs, timber, and CLT manufacturing sectors were set to 100% (import coefficient at 0). The import coefficient table defined the domestic self-sufficiency ratios of other sectors for 2015 [31].

*2.5. Evaluation of the Environmental Burden*

The embodied environmental burden intensity (EBI) based on producers' price was established as the amount of environmental burden generated directly and indirectly per unit of production activity (equivalent to JPY 1,000,000) in a sector. In other words, EBI is a coefficient that represents the total energy consumption and total GHG emissions per unit of production activity [32].

The input coefficients are the raw materials and fuel inputs required to produce one product unit in a sector and indicate the scale of raw materials and fuels utilized [28]. The input coefficients were calculated as follows:

$$a_{ij} = x_{ij}/X_j \tag{1}$$

$$x_i = \sum_{j=1}^{n} a_{ij}x_j + f_i \tag{2}$$

where $i$ denotes the number of row sectors, $j$ denotes the number of column sectors, the subscript represents the input of sector $i$ to sector $j$ to produce one unit of product, $x_{ij}$ denotes the input of sector $i$ to sector $j$, $X_j$ denotes the domestic output in sector $j$, $x_i$ and $x_j$ denote the gross output of sector $i$ and $j$, respectively, and $f_i$ denotes an increase in final demand in sector $i$ [28].

The inverse matrix coefficient table shows the expected final domestic production per sector when the final demand for one unit is produced for a specific sector [28]. We used the open inverse matrix coefficient B, implying that some economic or environmental impacts flowed from Japan. The open inverse matrix coefficients were calculated as follows:

$$B = (I - (I - \hat{M}) \times A)^{-1} \tag{3}$$

where I denotes the unit matrix, $\hat{M}$ denotes the diagonal matrix, zeros are non-diagonal elements, the import coefficients are diagonal elements, and $A = (a_{ij})$ denotes the input coefficient matrix.

By defining the output vector $x = (x_i)$, the final demand vector $f = (f_i)$, combining the definitions of Equation (3) and Equation (2), can be expressed as Equations (4) and (5):

$$x = (I - \hat{M}) \times Ax + f \tag{4}$$

$$x = (I - (I - \hat{M}) \times A)^{-1} f \tag{5}$$

"Total environmental burden" is the sum of the environmental burden generated by the increase in final demand, $f_i$, as specified by Equation (7). In this study, $d_i$ is referred to as the direct EBI of sector $i$, and is calculated by dividing the environmental

burden $D_i$, generated directly from sector $i$ by the total production value $x_i$, as shown in Equation (6) [32]:

$$d_i = D_i / x_i \tag{6}$$

$$E = \sum_{i=1}^{n} d_i x_i \tag{7}$$

By defining the direct EBI vector d = ($d_i$), the output vector x = ($x_i$), and the increase in the final demand vector f = ($f_i$), Equation (7) can be rewritten as Equation (8) [32]:

$$E = \begin{pmatrix} d_1 \\ \vdots \\ d_n \end{pmatrix}^t x = d^t \left( I - \left( I - \hat{M} \right) \times A \right)^{-1} f \tag{8}$$

The embodied EBI $e_k$ of sector $k$ denotes the sum of the environmental burden generated in each sector when the final demand of one unit is assigned to sector $k$. Therefore, Equation (9) can be calculated by defining the final demand vector $f_k$ (where $f_k$ is set to 1 and the other elements are set to 0) and substituting it in place of "f" in Equation (8) [32]:

$$e_k = d^t (I - (I - \hat{M}) \times A)^{-1} \begin{pmatrix} f_1 = 0 \\ \vdots \\ f_k = 1 \\ \vdots \\ f_n = 0 \end{pmatrix} = d^t (I - (I - \hat{M}) \times A)^{-1} f_k \tag{9}$$

Figure 2 shows the process of calculating the environmental burden. The expenditure on CLT manufacturing was defined as the increase in the final demand. An environmental burden calculation was performed.

The gross value-added (GVA) ratio was obtained by dividing the GVA of each sector by its corresponding domestic production value. The employee compensation ratio was obtained by dividing the employee revenue in each sector by the corresponding domestic production value. The consumption conversion ratio was calculated by dividing consumption expenditure by real income. Since the consumption conversion ratio varied annually, a 3-year moving average was adopted to calculate the consumption conversion ratio [33]. The consumption pattern was obtained by dividing the household consumption expenditure in each sector by the total household consumption expenditure.

The EBI was obtained from the 2015 3EID and the merging of 390 sectors into 187 to ensure economic and environmental data correspondence for the same sector [24,28]. Direct energy consumption and GHG emissions were calculated by multiplying the increase in final demand by EBI; indirect energy consumption and GHG emissions were obtained by multiplying the indirect production induced value by the EBI; and induced energy consumption and GHG emissions were obtained by multiplying the induced production value by the EBI. In the following analysis, the induced energy consumption and GHG emissions were included in the indirect energy and GHG emissions. The total energy consumption and GHG emissions were represented as the sum of the direct and indirect energy consumption and GHG emissions.

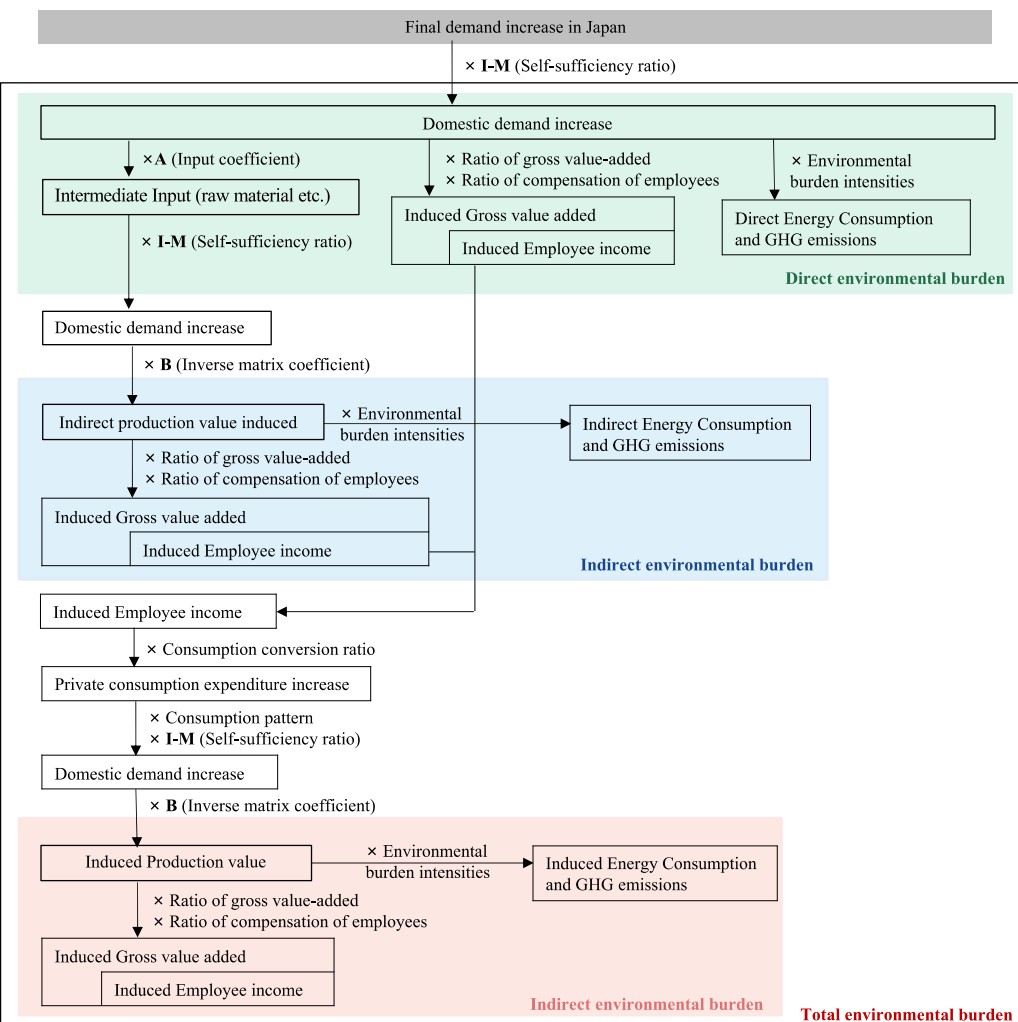

**Figure 2.** Flowchart of the environmental burden calculation.

## 3. Results

### 3.1. Environmental Burden of CLT Manufacturing

The environmental burden of the CLT manufacturing sector is listed in Table 2. The total energy consumption was 51,344 GJ, with an indirect consumption of 34,211 GJ (including an indirect consumption of 29,560 GJ and induced consumption of 4661 GJ) having the largest share, accounting for 67% of the total energy consumption. Total GHG emissions were 3336 t-$CO_2$eq, with indirect emissions at 2270 t-$CO_2$eq, accounting for 68% of the total GHG emissions, consistent with the share of energy consumption. Most of the GHG emissions originate from energy sources. At 3202 t-$CO_2$eq, they account for 96% of the total. The total CLT production in our preliminary study was 6399 m$^3$ [22]; thus, the average GHG emissions amounted to 0.521 t-$CO_2$eq/m$^3$.

Figure 3 shows the sectors where CLT manufacturing contributes to a high environmental burden. The industry generates more indirect GHG emissions along the supply chain than direct ones, and the environmental burden impacts a broader range of sectors. Direct GHG emissions were mainly from the use of electricity, activities not elsewhere classified, road freight transport (except self-transport), timber, and steam and hot water supply, with emissions of 665 t-$CO_2$eq, 186 t-$CO_2$eq, 142 t-$CO_2$eq, 45 t-$CO_2$eq, and 14 t-$CO_2$eq, respectively. Indirect GHG emissions were mainly from electricity consumption, road freight transport (except self-transport), activity not elsewhere classified, logs, and self-transport (freight), with emissions of 1161 t-$CO_2$eq, 224 t-$CO_2$eq, 188 t-$CO_2$eq, 131 t-$CO_2$eq, and 91 t-$CO_2$eq, respectively. Figure 4 shows the six sectors with the highest

total GHG emissions, namely electricity, an activity not elsewhere classified, road freight transport (except self-transport), timber, logs, and self-transport (freight), which accounted for 55%, 11%, 11%, 2%, 4%, and 3% of the total GHG emissions, respectively. The electricity sector had the most significant total GHG emissions of 1826 t-$CO_2$eq, accounting for 55% of the total GHG emissions, and indirect GHG emissions are approximately 1.7 times the direct GHG emissions.

**Table 2.** Environmental burden of the CLT manufacturing sector (values in brackets indicate induced consumption and emissions).

| | $CO_2$ (Energy Origin) (t-$CO_2$) | $CO_2$ (non-Energy Origin) (t-$CO_2$) | $CH_4$ (t-$CO_2$eq) | $N_2O$ (t-$CO_2$eq) | HFCs (t-$CO_2$eq) | PFCs (t-$CO_2$eq) | $SF_6$ (t-$CO_2$eq) | $NF_3$ (t-$CO_2$eq) | Total GHG Emission (t-$CO_2$eq) | Total Energy Consumption (GJ) |
|---|---|---|---|---|---|---|---|---|---|---|
| Direct | 1057 | 2 | 1 | 6 | 0 | 0 | 1 | 0 | 1066 | 17,123 |
| Indirect | 1859 (287) | 53 (11) | 8 (15) | 16 (8) | 10 (2) | 0 (0) | 1 (0) | 0 (0) | 1948 (322) | 29,560 (4661) |
| Total | 3202 | 66 | 24 | 30 | 12 | 1 | 2 | 0 | 3336 | 51,344 |

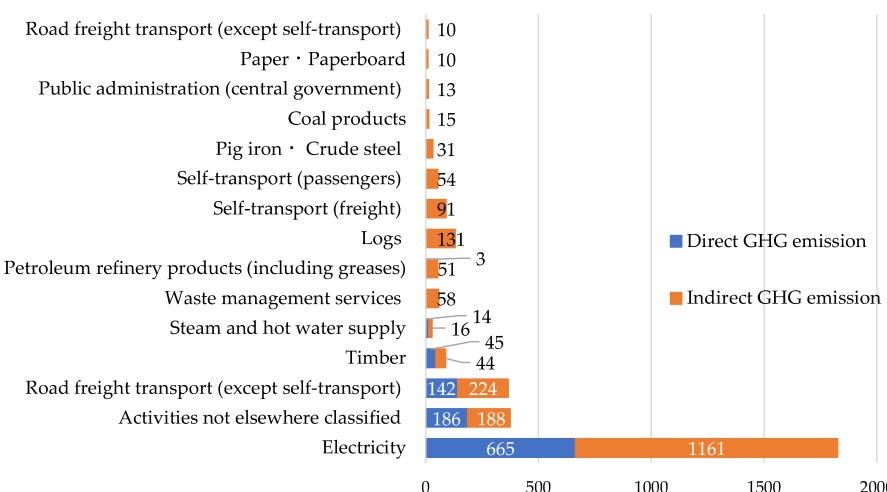

**Figure 3.** Sectors with high environmental burden of direct and indirect GHG emissions in the CLT manufacturing sector (unit: t-$CO_2$eq).

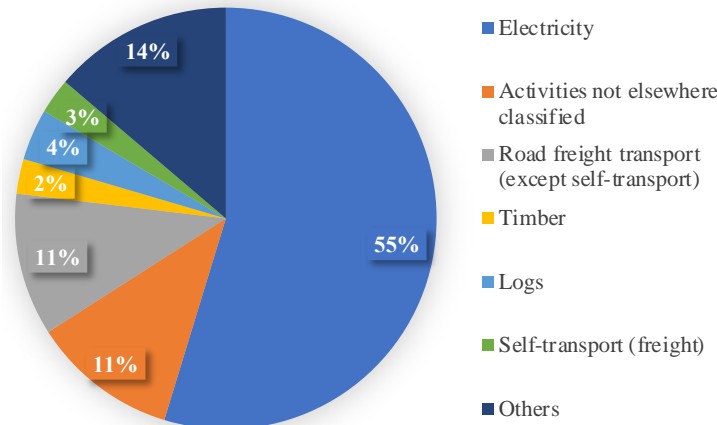

**Figure 4.** Top 6 contributors to the total amount of GHG emissions.

*3.2. Environmental Burden of Cement and Steel Sectors*

Under the same increase in the final demand (JPY 923,868,000) [22], the cement sector demonstrated a total energy consumption of 120,794 GJ, with direct consumption taking the largest share, at 66,522 GJ, accounting for 55% of the total (Table 3). Total GHG emissions were 19,975 t-$CO_2$eq, and direct emissions had the largest share of 74%

(=14,843 t-CO$_2$eq). Most GHG emissions originated from non-energy sources, accounting for 10,151 t-CO$_2$eq (51%).

**Table 3.** Environmental burden of the cement and cement product sector (values in brackets indicate induced consumption and emissions).

| | CO$_2$ (Energy Origin) (t-CO$_2$) | CO$_2$ (non-Energy Origin) (t-CO$_2$) | CH$_4$ (t-CO$_2$eq) | N$_2$O (t-CO$_2$eq) | HFCs (t-CO$_2$eq) | PFCs (t-CO$_2$eq) | SF$_6$ (t-CO$_2$eq) | NF$_3$ (t-CO$_2$eq) | Total GHG Emission (t-CO$_2$eq) | Total Energy Consumption (GJ) |
|---|---|---|---|---|---|---|---|---|---|---|
| Direct | 5845 | 8977 | 6 | 15 | 0 | 0 | 0 | 0 | 14,843 | 66,522 |
| Indirect | 3626 | 1160 | 10 | 25 | 12 | 0 | 2 | 0 | 4836 | 54,008 |
| | (10) | (13) | (7) | (2) | (0) | (0) | (0) | (0) | (196) | (263) |
| Total | 9481 | 10,151 | 23 | 42 | 12 | 1 | 2 | 0 | 19,975 | 120,794 |

Figure 5 shows sectors where cement production contributes a high environmental burden. Table 3 shows that the direct GHG emissions from the cement sector were 14,843 t-CO$_2$eq, 74%. Indirect emissions were predominantly from electricity, cement, cement products, self-transport (freight), pig iron and crude steel, road freight transport (except self-transport), and coastal and inland water transport, with emissions of 1996 t-CO$_2$eq, 1751 t-CO$_2$eq, 424 t-CO$_2$eq, 191 t-CO$_2$eq, and 146 t-CO$_2$eq, respectively. The top five sectors accounted for 93% of total indirect GHG emissions. As with the CLT manufacturing sector, electricity produced the largest GHG emissions.

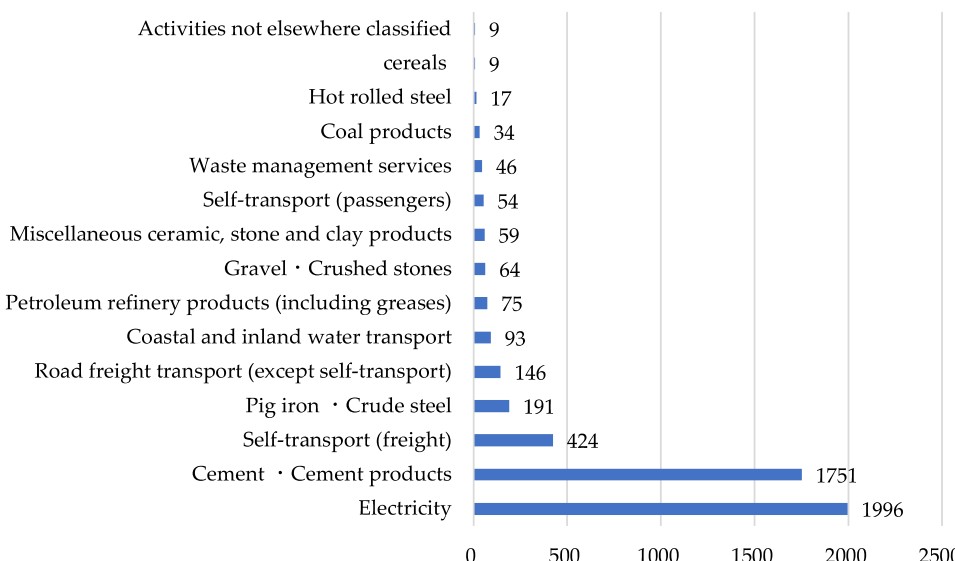

**Figure 5.** Sectors with high environmental burden of indirect GHG emissions in cement and cement product sector (unit: t-CO$_2$eq).

The environmental burden of the steel sector is shown in Table 4. It considered the same increase in final demand as in the CLT manufacturing sector (JPY 923,868,000) [22]. The total energy consumption was 222,008 GJ, with direct consumption taking the highest share of 52% (115,336 GJ). The total emissions amounted to 22,149 t-CO$_2$eq. Among those, direct emissions had the largest share, at 12,161 t-CO$_2$eq, which accounted for 55% of the total. Most of the GHG emissions were of the energy origin at 21,072 t-CO$_2$eq, which accounts for 95% of the total.

**Table 4.** Environmental burden of the steel sector (values in brackets indicate induced consumption and emissions).

| | $CO_2$ (Energy Origin) (t-$CO_2$) | $CO_2$ (Non-Energy Origin) (t-$CO_2$) | $CH_4$ (t-$CO_2$eq) | $N_2O$ (t-$CO_2$eq) | HFCs (t-$CO_2$eq) | PFCs (t-$CO_2$eq) | $SF_6$ (t-$CO_2$eq) | $NF_3$ (t-$CO_2$eq) | Total GHG Emission (t-$CO_2$eq) | Total Energy Consumption (GJ) |
|---|---|---|---|---|---|---|---|---|---|---|
| Direct | 11,571 | 556 | 11 | 12 | 0 | 0 | 0 | 0 | 12,161 | 115,293 |
| Indirect | 9451 (85) | 427 (3) | 16 (4) | 27 (2) | 4 (1) | 0 (0) | 2 (0) | 0 (0) | 9893 (95) | 105,336 (1379) |
| Total | 21,072 | 987 | 32 | 52 | 5 | 0 | 2 | 0 | 22,149 | 222,008 |

Figure 6 shows the sectors where steel production contributes to a high environmental burden. Table 4 shows that the direct GHG emissions from the cement sector were 12,161 t-$CO_2$eq, which is 55% of the total, thus following the description of only the indirectly impacted sectors. Indirect GHG emissions were concentrated in the pig iron, crude steel, electricity, and coal product sectors, with 6904 t-$CO_2$eq, 1735 t-$CO_2$eq, and 869 t-$CO_2$eq, respectively. The top three sectors accounted for 96% of total indirect GHG emissions. Unlike the CLT manufacturing and cement sectors, the steel sector is the largest contributor to GHG emissions, both directly and indirectly, at 86% of total GHG emissions, while electricity contributes only 8% of total GHG emissions.

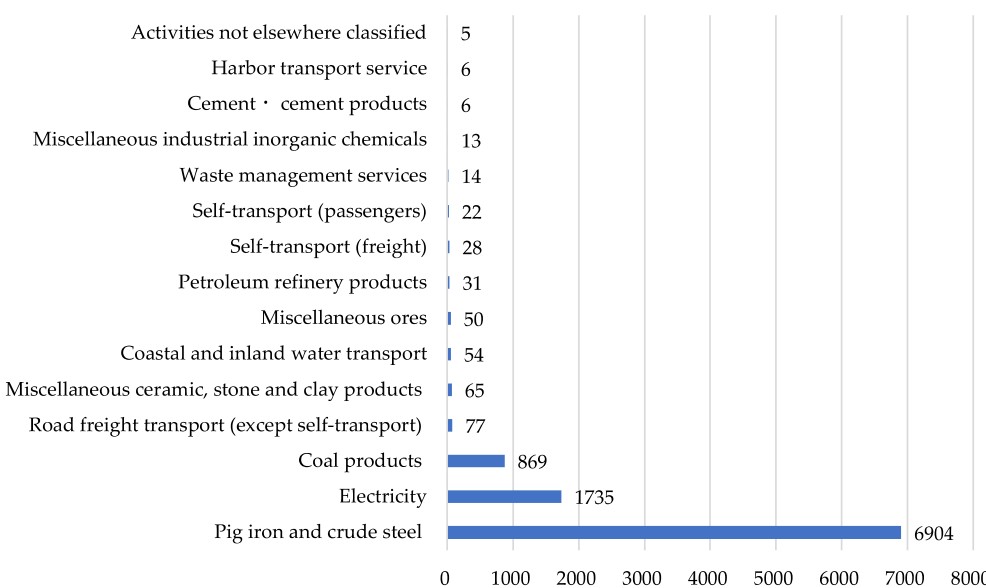

**Figure 6.** Sectors with the high environmental burden of indirect GHG emissions in the products sector in pig iron and crude steel sector (unit: t-$CO_2$eq).

## 4. Discussions

### 4.1. Analysis of the Results

The results of the environmental burden of the CLT manufacturing sector show that indirect production has the largest share of the environmental burden, with indirect emissions accounting for 68% of total GHG emissions. The primary emission sources come from electricity (55%), an activity not elsewhere classified (11%), and road freight transport (except self-transport) (11%) sectors. This suggests that producing raw materials (logs, timber, adhesives, etc.) and transporting materials along the chain of CLT manufacturing generates high GHG emissions. Meiken Lamwood Corp. uses biomass fuels to generate electricity to reduce GHG emissions. Since road freight transport (except self-transport) also accounts for 11% of total GHG emissions, shortening the distance for transportation of raw materials can also reduce GHG emissions to a large extent. In addition, $CO_2$ emissions of energy origin accounted for the largest share of each category of GHG emissions at 96%. $CO_2$ emissions of energy origin mainly come from the combustion of fossil fuels,

mainly used to provide electricity, which is in line with the result that the electricity sector produces the largest share of GHG emissions [34,35].

Comparing the results with the cement and steel sectors, the total GHG emissions from the CLT manufacturing sector were 3336 t-$CO_2$eq, approximately 17% those of the cement sector and 15% of emissions from the steel sector. The unit GHG emission for the CLT manufacturing sector was 3.61 t-$CO_2$eq/JPY one million, while the unit GHG emission of the cement and steel sectors were 21.62 t-$CO_2$eq/JPY one million and 23.89 t-$CO_2$eq/JPY one million, respectively. This suggests that the production of CLT to substitute cement and steel could reduce GHG emissions by 18.01 t-$CO_2$eq and 20.28 t-$CO_2$eq, respectively, for each one million JPY of domestic production generated. Compared with the CLT manufacturing sector, the cement and steel sectors have a large ratio of direct $CO_2$ emissions. Meanwhile, the cement sector has large direct $CO_2$ emissions of non-energy origin, whereas the steel sector has large direct $CO_2$ emissions of energy origin. The cement sector has a large non-energy origin because $CO_2$ is emitted from the thermal decomposition of limestone, the main raw material, whereas the steel sector is a typical energy-intensive industry.

Regarding sectors with a high environmental burden, electricity and transportation are still the main GHG emission sectors. Thus, using sustainable and clean energy resources to provide electricity, reducing transportation distances, and improving transportation efficiency are still important ways to reduce GHG emissions. Moreover, it is worth noting that the cement and steel sectors produce far more GHG emissions than the electricity sector; thus, applying CLT in medium- and high-rise buildings still has a significant market potential in terms of environmental burden.

*4.2. Comparison with Previous Studies*

We compared our results to those of other studies that used LCA. In our study, the average GHG emission from CLT manufacturing was 521 kg-$CO_2$eq/$m^3$ (including direct and indirect GHG emissions). Nakano et al. [20] assessed the environmental impacts of three CLT manufacturers in Japan using the LCA method. They estimated the GHG emissions from CLT manufacturing to be 252 kg-$CO_2$eq/$m^3$ [20]. The difference in the results may be owing to different system boundaries. Nakano et al. used the LCA method to focus on the production of CLT products, whereas in this study, we used input–output analysis covering the production of CLT up to its use in construction. In the input–output calculation, we used all the outputs of the CLT manufacturing sector in the non-residential and residential sectors, and the outputs to the non-residential and residential sectors indirectly increased their GHG emissions. In addition, among the manufacturing processes, the use of electricity in laminae production had the highest GHG emissions, which is consistent with the findings of this study that electricity was the main GHG emission source [20]. Chen et al. [21] and Puettman et al. [36] also used process analysis in the LCA method to study the CLT production in western Washington and Oregon, USA, and found that they emit 156 kg-$CO_2$eq/$m^3$ and 142 kg-$CO_2$eq/$m^3$, respectively. Process analysis is generally considered more accurate; however, for accuracy, it has to cover the entire life cycle. This accuracy is affected when some parts of the life cycle are excluded, as in Chen et al. [21] and Puettman et al. [36], where the GHG emissions from transportation were excluded. Input–output analysis can set a broader system boundary, such as the input–output table, and, therefore, has the potential to cover a broader life cycle than process analysis. In addition, indirect GHG emissions in this study accounted for a large proportion (58%) of the total GHG emissions. If only direct GHG emissions were considered, the average GHG emissions would have been 166 kg-$CO_2$eq/$m^3$. This result is closer to those of the mentioned studies, illustrating that input–output analysis covers a broader life cycle.

## 5. Conclusions

This study quantified the environmental burden of CLT manufacturing based on the results of preliminary studies on the economic ripple effects of CLT manufacturing in combination with energy and emission intensity data. In addition, the environmental

burdens of the cement and steel sectors were evaluated for comparison with the same increase in the final demand. The key findings in this study are as follows:

With an increase in the final demand of JPY 923,868,000, the total energy consumption of CLT manufacturing was 51,344 GJ, total GHG emissions were 3336 t-$CO_2$eq, and average GHG emissions were 0.521 t-$CO_2$eq/$m^3$. The environmental burden of manufacturing CLTs is concentrated in electricity, activities not elsewhere classified, and road freight transport (except self-transport). The results indicate that using renewable energy for electricity generation and the proximity transportation of raw materials are important ways to reduce GHG emissions.

For the same increase in final demand, the cement and steel sectors have a much more significant environmental burden than the CLT manufacturing sector, accounting for approximately 17% and 15% of the GHG emissions and steel sectors' GHG emissions, respectively. The production of CLT instead of cement and steel could reduce GHG emissions by 18.01 t-$CO_2$eq and 20.28 t-$CO_2$eq, respectively, for each million JPY generated domestic production. In addition to the fact that electricity accounts for the major GHG emissions in all three sectors, cement and steel production has a much larger environmental burden than that of the electricity sector. Therefore, regarding the environmental burden, there is still an immense potential for applying CLT in medium- and high-rise buildings.

The results of various CLT environmental burden studies may vary depending on the timing of the investigation and database. A limitation of this study is the focus on only one CLT manufacturer. Because CLT is still a relatively new structural material in Japan, we can consider a more extensive and systematic environmental burden study of CLT, for example, by surveying eight CLT manufacturers throughout Japan to evaluate the overall environmental burden of CLT manufacturing or predicting the economic and environmental impacts of reaching government-targeted prices and production volumes. In addition, the methodology of using extended input–output in this study can be broadly applied to the environmental burden studies of other specific industries and their supply chain. In addition, based on the results of larger economic ripple effects and smaller environmental burdens of CLT manufacturing, CLT still has great potential for development in construction and civil engineering.

**Author Contributions:** Conceptualization, M.L. and C.K.; methodology, M.L. and C.K.; data curation, M.L., A.M. and C.K.; investigation, M.L., A.M. and C.K.; formal analysis, M.L.; writing—original draft preparation, M.L.; writing—review and editing, C.S.G. and C.K. All authors have read and agreed to the published version of the manuscript.

**Funding:** This research was funded by the Japan Society for the Promotion of Science, grant number JP23H03596; Institute of Global Innovation Research in Tokyo University of Agriculture and Technology; and FLOuRISH JIRITSU Fellowship in Tokyo University of Agriculture and Technology.

**Data Availability Statement:** The data presented in this study are available on request from the corresponding author.

**Acknowledgments:** We are deeply grateful to the Meiken Lamwood Corp. for providing invaluable data and assistance for this study.

**Conflicts of Interest:** The authors declare no conflict of interest. The funders had no role in the design of this study, in the collection, analyses, or interpretation of data, in the writing of the manuscript, or in the decision to publish the results.

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
