# Peer review of "Evaluation of the Environmental Burden of Cross-Laminated Timber Manufacturing in Japan Using the Input–Output Analysis"

_forests, doi:10.3390/f14112263_

Round 1
Reviewer 1 Report
Comments and Suggestions for Authors
The paper evaluating the environmental burden of cross-laminated timber (CLT) manufacturing in Japan makes a compelling case for the environmental benefits of CLT manufacturing compared to cement and steel sectors. The manuscript smartly pinpoints the main elements contributing to emissions from CLT manufacturing. By highlighting identifiable sectors like electricity, road freight transportation, and timber and logs, the paper demonstrates a detailed understanding of the factors affecting the carbon footprint of CLT manufacturing. However, the range of sources cited suggests that the authors undertook a comprehensive review of existing literature on CLT and its impacts. The manuscript is generally good and seems suitable for publication in the journal forests. I add a few suggestions below for the authors to improve the manuscript further.
Line 51: I wonder if a definition of CLT could be added at the beginning for readers who are far from the subject? Just a suggestion.
Line 24, line 80, line 245, line 249 etc. “Activities not elsewhere classified” What are these? Can you give some details? This sentence makes no sense to the reader.
Line 320- 333: The environmental burden of CLT is good compared to the environmental burdens of the cement and steel sectors. OK, there is nothing wrong with that. But how does it compare to different types of wood composites? Is CLT production advantageous? Can a comparison be made on this? Are there any such findings in previous studies?
Line 368: The conclusion section summarizes the findings but does not provide clear recommendations or implications for future research or policy. Adding a more robust conclusion with actionable takeaways would enhance the manuscript’s practical value.
Reviewer 2 Report
Comments and Suggestions for Authors
The paper is interesting and probably important. This reviewer has read the paper, but gained a limited understanding. This is possibly due to the limited experience of this reviewer in Life-Cycle Analysis and econometrics. Correspondingly, the Editor may mostly rely on the statements of other reviewers.
However, some comments are here provided, in the hope that they might be useful for the Authors.
The Environmental Burden Intensity is based on total energy consumption and total GHG emissions per unit of production activity. Does it matter whether the energy resources are renewable or non-renewable? Does it matter whether the GHG emissions are compensated by carbon capture processes somewhere in the production chain (by growing trees, or the like)?
This reviewer has difficulties in following the abstract mathematics on page 6.
